# Ultracentrifugation Techniques for the Ordering of Nanoparticles

**DOI:** 10.3390/nano11020333

**Published:** 2021-01-27

**Authors:** Xufeng Xu, Helmut Cölfen

**Affiliations:** 1Laboratory of Physical Chemistry, Department of Chemical Engineering and Chemistry, Eindhoven University of Technology, 5612AE Eindhoven, The Netherlands; x.xu3@tue.nl; 2Physical Chemistry, University of Konstanz, Universitätsstraße 10, Box 714, 78457 Konstanz, Germany

**Keywords:** centrifugation, sedimentation, nanoparticle, concentration gradient, non-equilibrium process, superstructure, functional material

## Abstract

A centrifugal field can provide an external force for the ordering of nanoparticles. Especially with the knowledge from in-situ characterization by analytical (ultra)centrifugation, nanoparticle ordering can be rationally realized in preparative (ultra)centrifugation. This review summarizes the work back to the 1990s, where intuitive use of centrifugation was achieved for the fabrication of colloidal crystals to the very recent work where analytical (ultra)centrifugation is employed to tailor-make concentration gradients for advanced materials. This review is divided into three main parts. In the introduction part, the history of ordering microbeads in gravity is discussed and with the size of particles reduced to nanometers, a centrifugal field is necessary. In the next part, the research on the ordering of nanoparticles in analytical and preparative centrifugation in recent decades is described. In the last part, the applications of the functional materials, fabricated from centrifugation-induced nanoparticle superstructures are briefly discussed.

## 1. Introduction

### 1.1. The Era of Microbeads in Gravity

The sedimentation of particles is a long-standing phenomenon in nature. One of the most impressive examples may be the formation of opals [1,2,3,4] where the sedimentation of silica particles gradually forms iridescent mineraloids over a long period of time. Besides these great phenomena in nature, the first experimental work to study particle sedimentation was conducted by Jean Perrin [5] in 1912. He used an optical microscope to observe the 3-dimensional motions of microscopic gamboge particles during their sedimentation, under gravity. The observation directly proved the constant random motion of any small object, was raised by Albert Einstein [6] in 1905 and also led to an experimental estimation of the Boltzmann constant. This work paved the way to the understanding of the reality of atoms and molecules, and thus earned Jean Perrin the Nobel Prize for Physics in 1926. After this breakthrough work, extensive research on the movement of particles in sedimentation has been conducted from the early-20th century on, especially the establishment of fundamental theories of sedimentation [7,8,9,10,11]. Besides these theoretical advances, experimental research also saw major progress [12,13,14,15,16], especially the classical experiments conducted by P.N. Pusey and W. van Megen. They synthesized hard-sphere latex particles, which represent the simplest possible interparticle interaction [17] case. They observed the ordering process of these microbeads with a good size monodispersity (diameter: 0.61 µm ± 0.02 µm) under gravity and the final formation of Bragg-reflecting crystalline samples [18], which demonstrated an iridescent structural color [19,20] in front of a beam of white light, as shown in Figure 1. The phase diagram was also obtained, which described the ordering transition from fluid to crystal [21,22] and then to glass with an increasing volume fraction. Research on the ordering of binary hard sphere mixtures in dispersion followed [23,24], which displays a richer phase diagram [2] with a large variety of ordered structures, as shown in Figure 2. With these exciting experimental observations, the theoretical studies also stepped forward [25,26], particularly in the area of fundamental studies on crystal nucleation and growth [27,28] of particles. At the beginning of the 21st century, the idea of a combination of fluorescence labelling [29] and refractive index matching was realized by van Blaaderen et al. and thus allowed in-situ observation [30,31,32] of the whole sedimentation and ordering process of colloidal particles in a gravitational field. The representative examples [33,34] are displayed in Figure 3. These studies led to the visualization of the whole ordering process of the particles during sedimentation and provide insights into the tuning [35,36] of the ordering of superlattice structures, typically for near-micrometer-sized colloids.

### 1.2. The Era of Nanoparticles: Gravity Is Not Sufficient Anymore

Nanotechnology [37,38,39] has been developing rapidly since its emergence in the 1980s, especially in the field of nanoparticles. A large variety of nanoparticles (with the diameter typically between 1 and 100 nm in at least one dimension [40]) can be synthesized by well-established protocols, such as quantum dots [41], silica [42], and noble metal nanoparticles [43]. These nanoparticles can be furthermore ordered into superstructures via so-called bottom-up organization [44]. The important examples are colloidal crystals made of spherical nanoparticles [45] as well as mesocrystals [46,47,48] made from anisotropic nanocrystals, as shown in Figure 4. These superstructures are proven to be promising as functional materials in many different applications [49,50,51,52,53,54]. The preparation approaches for these superstructures are reviewed in several key papers [55,56,57]. Especially in the review paper by Boles et al. [55], all widely used methods are summarized, as shown in Figure 5. Among these methods, solvent evaporation and destabilization are the two most popular methods, with a wide selection of examples [58,59,60]. However, the traditional gravitational sedimentation approach, which worked quite well for near-micrometer-sized colloids, becomes a much less commonly used method for nanoparticles due to the lack of effective sedimentation for particles <1 μm under gravity, consequently resulting in very limited research [61]. The plausible hurdles may be two-fold. Firstly, it is the intrinsically small size of nanoparticles that makes the sedimentation extremely slow under gravity. To elaborate this in a quantitative manner [62], the gravitational energy to move a particle with the mass *m* by its own diameter, *d* can be calculated: *mgd* (*g* is the Earth’s gravity resp. gravitational acceleration). The significance of the gravity to the sedimentation of particles can be calculated: *mgd*/*k*B*T* with *k*B*T* being the thermal energy scale. For microbeads with a typical diameter of 1 µm, the value is in the order of 106, which means that the gravity is so significant compared to thermal motion that sedimentation progresses in a relatively reasonable time scale. In comparison, for nanoparticles with a diameter of 100 nm, the value is significantly decreased to 10^−6^. This means that the gravity becomes so insignificant that nanoparticles can hardly sediment in a gravitational field but move by diffusion caused by particle collisions. Thus, an external centrifugal field [63] is compulsory for the sedimentation of nanoparticles instead of natural sedimentation. The second hurdle is the lack of in-situ analytics to monitor the sedimentation of nanoparticles during centrifugation. For example, confocal laser scanning microscopy [34,64,65] (CLSM), which worked perfectly with microbeads, does not have enough resolution to observe nanoparticles in-situ. Therefore, the understanding of the ordering process for nanoparticles is still missing, not to mention the significant turbidity of the dispersion at very high nanoparticle concentration, especially close to the fluid to crystal transition point (at around 50 vol%), which severely hinders the in-situ observation. These main problems may have caused the sedimentation method to be unpopular so far for the ordering of nanoparticles, even though it is quite intuitive, convenient, and can be easily scaled-up.

### 1.3. Ordering of Nanoparticles: Centrifugation Needs to Be Involved

Recent decades have seen the development in both analytical (ultra)centrifugation (AUC) [66] and preparative [67] (ultra)centrifugation (PUC) techniques, with important software and hardware advances [68,69,70,71,72]. The application of this knowledge may make the ordering of nanoparticles more accessible in the centrifugal field, finally reaching the goal of tailor-made superstructures by centrifugation. It is worthwhile mentioning that one key advantage of using centrifugation is the possibility of a continuous variation of the force exerted on the nanoparticles with the distance to the rotational center and thus the formation of concentration gradients. This will be extremely useful for the ordering of nanoparticles as richer superstructures will be possible in this out-of-equilibrium gradient [57,73] rather than a homogenously distributed suspension. In this review, we will focus on the ordering of nanoparticles in a centrifugal field, both in analytical and preparative instruments. Both the intuitive use of centrifugation for nanoparticle ordering and more rational research work with the assistance from powerful analytical (ultra)centrifugation in recent years will be summarized. At the end of the review, the current problems and the future possibilities in this field will be discussed.

## 2. Ordering of Nanoparticles in Analytical (Ultra)Centrifugation (AUC)

### 2.1. General Introduction to AUC

Analytical (ultra)centrifugation (AUC) is an important first principles technique to precisely characterize colloidal nanoparticles in dispersion in real-time [68]. The first ever AUC experiments can be traced back to the work [11,74,75] done by T. Svedberg and his co-workers in the 1920s. They demonstrated the ability of AUC for the determination of the size and size distribution of nanoparticles, such as gold nanoparticles with the diameter of only 2 nm, in dispersion, even before the invention of advanced electron microscopy [76]. AUC was also found to be a very powerful characterization method for (bio)macromolecules already in the very early days, starting with the pioneering work on hemoglobin proving its macromolecular character [77]. As a consequence, the AUC technique became widely used in the field of biochemistry and molecular biology, especially for the analysis of biological macromolecules such as proteins, nucleic acids and so on [78,79]. However, even after decades this technique is still powerful for nanoparticles and new methodologies are still being developed [71]. Particularly, the ordering process of nanoparticles in a centrifugal field can be measured in-situ and controlled in AUC, which can thus provide extremely useful knowledge for tailor-made superstructures in preparative (ultra)centrifugation (PUC). In a typical AUC experiment, dissolved or dispersed samples inside an AUC measuring cell are exposed to a centrifugal field induced by the spinning of a centrifuge rotor. The reaction of the sample to this centrifugal field is in-situ detected by optical detection systems in real time [68]. A schematic setup of a Beckman CoulterTM AUC instrument (with an UV absorbance detector system) is illustrated in Figure 6A. In terms of optical detection systems, AUC can be classified into five basic types: Schlieren optics, Interference optics, UV-vis absorbance optics, Fluorescence optics and Turbidity optics [80]. In recent years, another advanced optical detector system was developed, that is the multi-wavelength (MWL) optics [81,82,83,84,85]. This detector system allows for the simultaneous measurement of multiple species by producing a three-dimensional information set (absorbance × wavelength × radial position), as shown in Figure 6B.

The two important theoretical equations in the field of AUC are the Svedberg Equation and the Lamm Equation. In deriving the Svedberg equation, three forces are assumed to come into balance immediately in a centrifugal field, including sedimentation (*F*_s_) and the counter acting buoyancy (*F*_b_) and frictional (*F*_f_) force which thus induce the sample species to move with a constant sedimentation velocity, as shown in Equation (1). With the expansion of these three forces and several rearrangements, the famous Svedberg Equation can be deduced, as shown in Equation (2) and its derivation for hard spheres in Equation (3) [68].
(1)$$F_{s}+F_{b}+F_{f}=0$$
(2)$$M=\frac{sRT}{D\left(1-\bar{\nu }\rho_{s}\right)}$$
(3)$$d_{H}=\sqrt{\frac{18\eta_{s}s}{\left(\rho_{p}-\rho_{s}\right)\;}}$$
where *M* is the molecular mass of the species, *s* is the sedimentation coefficient, *R* is the molar gas constant, *T* is the thermodynamic temperature, *D* is the diffusion coefficient, *ρ*_p_ and *ρ*_s_ are the density of the species and solvent respectively, $\bar{\nu }$ is the partial specific volume, *d*_H_ is the hydrodynamic diameter and *η*_s_ is the solvent viscosity.

In comparison, the general Lamm Equation is deduced by a more accurate thermodynamic approach [10,68]. It describes the sample concentration *c* change with time *t* and radius *r* during the centrifugation at an angular velocity *ω*, as shown in Equation (4) and it basically holds for all kinds of AUC experiments.
(4)$$\frac{\partial c}{\partial t}=D\left(\frac{\partial^{2}c}{\partial r^{2}}+\frac{1}{r}\frac{\partial c}{\partial t}\right)-\omega^{2}s(r\frac{\partial c}{\partial r}+2c)$$

The two main AUC experiments, which can be used for the ordering of nanoparticles, are the sedimentation velocity (AUC-SV) and the sedimentation equilibrium (AUC-SE) experiments. A typical AUC-SV experiment is carried out in a strong centrifugal field, which allows for the complete sedimentation of the sample. By applying Equations (2) and (3), the molecular mass and size distribution can be thus obtained for the measured samples. In comparison, a typical AUC-SE experiment is carried out in a moderate or low centrifugal field. Therefore, the sedimentation process is balanced by a back-diffusion process for the sample and finally an equilibrium state can be achieved. This means *∂c*/∂*t* = 0 in Equation (4) and a concentration gradient *c*(*r*) along the radius *r* is established accordingly to the equilibrium state. It is also worthwhile to mention that the final concentration gradient (at equilibrium) is determined by thermodynamics, such as the absolute molar mass and the second virial coefficient/interparticle interaction of the investigated sample. Therefore, we can understand that AUC-SE experiments are far more important for the nanoparticle ordering since a concentration gradient can be formed and controlled in this process. In the following sections, the applications of AUC-SV and AUC-SE for the ordering of nanoparticles in a centrifugal field will be discussed.

### 2.2. Ordering of Nanoparticles in AUC-SV

In a typical AUC-SV experiment, a strong centrifugal field is normally applied to induce a complete sedimentation of nanoparticles. From the full sedimentation process, the sedimentation coefficient distribution of the investigated nanoparticles can be measured precisely. Therefore, the traditional aim of an AUC-SV experiment is to determine the size distribution of nanoparticles, typically by using Equation (3). Two representative examples are given by Carney et al. [87] and Karabudak et al. [82] Carney et al. demonstrate that by extracting the sedimentation and diffusion coefficient distributions from an AUC-SV experiment, accurate molecular weight and density distributions can be calculated in addition to the size distribution for ultra-small (~10 nm) core-shell nanoparticles, which solved a complicated and common problem since synthetic inorganic-core nanoparticles are mostly covered by organic capping agents, which have a different density and molecular weight. Karabudak et al. show that a sub-nanometer resolution can be reached for the size distribution of CdTe quantum dot mixtures with the power of a novel multi-wavelength detector [72,81,83,84,85,88]. These two cases demonstrate the traditional applications of an AUC-SV experiment. It can yield the accurate determination of the size distribution and size polydispersity of nanoparticles. These examples show that AUC-SV experiments can provide the size and size polydispersity of nanoparticles in unprecedented accuracy, which is crucial for the ordering of nanoparticles [89,90]. Moreover, AUC-SV experiments provide the sedimentation distribution of a sample, which can thus guide the precise separation/purification [91]. This strategy may be used to separate different colloidal superstructures [92] of both isotropic [93] and anisotropic building blocks [94,95].

A novel application of AUC-SV experiments has recently been introduced by Spinnrock and Cölfen. In their study, nanoparticles were ordered into a gradient superstructure in a centrifugal field in a tailored way by simulating concentration gradients using the common AUC theory [70,96]. The idea originated from the controlled preparation of gradient polymers [97] by in-situ forming a macromolecular initiator concentration gradient in an AUC experiment. The idea of this work was then applied to the in-situ formation of nanoparticle concentration gradients in real time [98]. Spinnrock et al. ordered spherical gold nanoparticles (with the diameter of 23 nm) in liquid gelatin as a solvent into a concentration gradient during the sedimentation process at 40 °C and then fixed the desired concentration gradient by solidification of gelatin upon cooling down to room temperature, as shown in Figure 7A. The desired gradient can be calculated and designed beforehand, using AUC theory implemented in common AUC evaluation programs like SEDFIT [99] or UltraScan [100], which agreed very well with the experiments, as shown in Figure 7B. This approach was also proven to be versatile, as the gradient superstructures can be applicable to nanoparticles of different sizes, shapes and materials. More interestingly, nanoparticle gradient materials can also be fabricated in an AUC band-forming cell, by the diffusion process at a very low angular velocity in an AUC synthetic boundary experiment [101], as shown in Figure 7C. In this experiment, the nanoparticles were overlayed onto a solution column of molten gelatin at around 40 °C at a low angular velocity. Afterwards, the nanoparticle layer continued to diffuse into the solvent column, which can be detected by AUC in real-time. After the desired concentration gradient was reached, the polymer solution was immediately cooled down to room temperature, thus fixing the gradient. Moreover, it was shown that the gradient can be simulated beforehand after the diffusion coefficient was extracted using Fick’s second law of diffusion. These examples show that AUC-SV experiments also find novel applications for directly ordering nanoparticles into tailor-made gradient superstructures.

### 2.3. Ordering of Nanoparticles in AUC-SE

A series of tailor-made nanoparticle gradient superstructures can be accessed via AUC-SV experiments in a thermo-reversible gel matrix, but there are some remaining challenges, especially the effect of convection [96]. Since the concentration gradient is fixed during the dynamic sedimentation process, the solvent back-diffusion, temperature, and density gradients might cause convection and lead to disturbance in the concentration gradient, especially for large volumes in large-scale practical applications. Another minor issue is the steep slope of the sigmoidal concentration gradient, particularly for very monodisperse nanoparticles. It is not convenient to broaden it by diffusion at low speeds [102] if a much more expanded nanoparticle gradient is desired, because diffusion processes are slow. However, these two problems can be easily avoided if AUC-SE experiments are employed, since in an AUC-SE experiment, any transport process is absent and the slope of the gradient can be controlled easily, for example, by changing the particle size or angular velocity. Therefore, a more controllable nanoparticle gradient can be established in an AUC-SE experiment, as reported recently by Xu and Cölfen [103]. In their study, polyethylene glycol (PEG) stabilized silica nanoparticles with the diameters of around 30 and 40 nm were synthesized, which served as good hard sphere models [104]. In this case, any complicated interparticle interaction other than the excluded volume interaction was avoided. The other important step in the study is to use a glycerol and water mixture (80 vol% glycerol + 20 vol% water) for refractive index matching of the silica nanoparticles (*n* ≈ 1.45) so that the concentration gradient of fluorescence labelled silica nanoparticles can be in-situ measured to a good accuracy even at an extremely high particle concentration, approaching the closest packing volume fraction (~70 vol%). For multiple nanoparticles, different fluorescence labels [105] are required with the assistance of an MWL detector. Finally, it is demonstrated that the concentration gradients of monodisperse and binary nanoparticles can be in-situ measured by AUC, irrespective of initial concentrations and sample volumes. More importantly, these concentration gradients can be even simulated in SEDFIT with the input knowledge of sedimentation, diffusion and nonideality coefficients, as shown in Figure 8A,B. The experimental and simulation results agreed very well in these cases. This allows transfer of the results to the preparative scale since the concentration gradients can be simulated for any ultracentrifuge rotor if the dimensions are known. These concentration gradients can be further tailor-made by tuning particle sizes and concentration ratios to build desired nanoparticle gradient materials, which displayed different superstructures at different radial positions [106,107]. It is also noteworthy that in the ordering process, smaller nanoparticles may float and form a more flattened concentration gradient due to the effective solvent density [106], as shown in Figure 8C. An effective solvent density rather than a constant solvent density needs to be applied when the concentration gradients are simulated in a multiple particle system [108,109], especially if the particle size ratio is large. A more complicated but intriguing scenario may occur when these nanoparticles are charged, due to additional electrostatic interactions and the presence of a large amount of counter-ions [110,111,112,113,114]. Rasa and Philipse showed that charged colloids are ordered in an much more extended concentration gradient due to the Donnan potential [115,116,117,118], which may even lead to a three-region concentration profile. Region I and III can be fitted by barometric equations while Region II follows a quadratic equation. Therefore, for charged nanoparticles, more parameters are available to tune the concentration gradients, especially the particle charge and solvent ionic strength, which makes the concentration gradient much more complicated. Accordingly, much richer superstructure phases are expected to be fabricated [107] for charged nanoparticles in the sedimentation-diffusion equilibrium.

### 2.4. Transition from AUC to PUC

Overall, it has been demonstrated that both sedimentation velocity (AUC-SV) and sedimentation–diffusion equilibrium (AUC-SE) experiments can be employed to order nanoparticles into concentration gradients and AUC-SE may be better in the sense that any transport is avoided and the concentration gradient is already “fixed” in the equilibrium state. However, AUC cells are initially designed for analytical purposes so that the cell volume is very limited (with the maximum value up to several hundred microliters), as compared to a centrifugal tube (several milliliters which is more than ten times larger) in preparative (ultra)centrifugation (PUC) experiments. Typical scales of nanoparticle gradient materials, made by these two centrifugation techniques, are compared (0.25 cm^3^ vs. 5.5 cm^3^), as shown in Figure 9A,B. Therefore, whether the knowledge of nanoparticle ordering in AUC experiments can be applied to PUC experiments is a crucial issue, especially if we aim to make this methodology more practical and even useful for industrial applications. In this case, proof-of-concept experiments were conducted to demonstrate clearly that the AUC experiments can be scaled up to PUC experiments, as shown in Figure 9C,D. The simulations still worked well for the nanoparticle concentration gradients made in PUC. In the following chapters, the ordering of nanoparticles in PUC experiments will be discussed, which has a larger impact on functional material fabrication and applications.

## 3. Ordering of Nanoparticles in Preparative (Ultra)Centrifugation (PUC)

### 3.1. General Introduction to PUC

Preparative (ultra)centrifugation (PUC) is a widely used method in a variety of disciplines, especially in chemistry and biology. It is employed to apply an enhanced gravity effect up to 100,000 rpm (correspondingly roughly 800,000× *g*), which is suitable for the sedimentation of samples of a wide size range from micron-sized species down to the smallest nanoparticles [119,120]. Its conventional application relies on the isolation/purification of a mixture of species. The mechanism is that different species have varied sedimentation velocities due to their intrinsic properties such as size, density and shape. The main centrifugation-assisted separation methods [121,122,123] include differential centrifugation, isopycnic separation and density gradient centrifugation [124]. The rotors for PUC can be divided into four main categories: swinging-bucket, fixed-angle, vertical and near-vertical, as shown in Figure 10. In recent decades, PUC has demonstrated its advantage in nanoparticle purification. Gold nanoparticle mixtures of different shapes [125,126] and sizes [127], graphene oxide [128] and carbon nanotube mixtures [129] of different lengths were all demonstrated to be purified quite well by PUC, especially with the pre-knowledge from AUC experiments. In this chapter, we focus, however, on the ordering of nanoparticles in PUC, which has also shown many promising results in recent decades. Especially with the assistance of AUC, the PUC experiments become more rational, inducing fully controllable ordered superstructures.

### 3.2. Ordering of Monodisperse Nanoparticles in PUC

The ordering of monodisperse nanoparticles in PUC was demonstrated in the 1990s by several groups [130,131,132,133,134,135,136,137,138] to be a quick, robust and versatile approach for the formation of three-dimensional crystalline superstructures, which can be further fabricated into microporous materials. A representative example was shown by Holland et al. [132,133]. In their study, monodisperse latex nanoparticles with a very narrow size distribution were firstly synthesized. Then a dispersion of these latex nanoparticles was centrifuged into a sediment (which showed an iridescent color) at the bottom of a centrifuge tube. Finally, the sediment was observed by scanning electron microscopy (SEM), to be a close-packed array of latex nanoparticles, as shown in Figure 11A. This piece of three-dimensional close-packed crystal can be further used as a template for a variety of inorganic or hybrid porous structures, particularly of titania [131] (TiO_2_) and silicon [136] (Si) which both have a very high refractive index and are thus very promising for photonic applications [139]. Besides three-dimensional structures, monodisperse silica nanoparticles were demonstrated by Fan et al. [140] to order into two-dimensional crystalline films by centrifugation, as shown in Figure 11B. During centrifugation, the aqueous phase was forced to float with the sinking of the hydrocarbon mixture phase (as less dense hexane in the mixture evaporated). In this case, the silica nanoparticles were separated from the aqueous phase and ended up in the interface between the hydrocarbon mixture and oil phases. At the interface, these nanoparticles were finally ordered into a crystalline film due to interparticle capillary water bridges. In addition to crystalline superstructures, nanoparticles were also demonstrated to be ordered into glassy superstructures [141,142,143] in the PUC. Garcia et al. [141] found the two methods, namely the destabilization of charge-stabilized nanoparticles and the introduction of nanoparticles of a different size, to make colloidal glasses, as shown in Figure 11C–E. Recently, it was shown by Chen et al. [144] that by the use of AUC as an in-situ characterization technique, the amount of oligomers formed during the centrifugation could even be quantified in nanoparticle dispersions with different salt concentrations. Therefore, the nanoparticle ordering can be controlled rationally in the PUC by varying salt concentrations, as shown in Figure 12. The final example is the research work presented by Roca et al. [145] They found that by using the PUC, gold nanoparticles can be ordered into clusters of different sizes. Moreover, the composition of the gold nanoparticle clusters can be controlled by tuning the angular velocity in the PUC. To conclude, we showed that the ordering of monodisperse nanoparticles in the PUC began with intuitive centrifugation experiments, which proved robust in the ordering of nanoparticles into both crystalline and glassy superstructures. Nevertheless, with the assistance of AUC techniques as an in-situ characterization method, the PUC experiments can become more rational and controllable, with the ordering of monodisperse nanoparticles fully tunable between crystals and glasses.

### 3.3. Ordering of Binary Nanoparticles in PUC

The ordering of binary nanoparticles in PUC is much more complicated compared to the ordering of one species of nanoparticles, because much more parameters play vital roles in the ordering process, including the particle size/number/charge ratio, the total particle concentration, the centrifugation time, the centrifugal force and so on [146]. Therefore, an intuitive PUC experiment is even impossible for binary nanoparticles, which otherwise worked very well for the ordering of the single species cases. Only recently, Chen et al. [107] showed that with the important sedimentation coefficient information obtained from AUC experiments, the rational design of binary nanoparticle ordering in PUC became possible. In their study, they first evaluated the sedimentation coefficient distributions, which are dependent on the total volume fraction, for the binary latex nanoparticles by using the AUC. With this information, they designed the PUC experiments with different conditions by varying total particle volume fractions, number ratios, total volumes, centrifugation time and centrifugal force. They finally found that binary nanoparticles ordered into different superstructures in the PUC experiments, which showed dependence on different radial positions in a centrifugation tube, as shown in Figure 13A. The main reason for this is the formation of different concentration gradients of the binary nanoparticles by centrifugation. Xu et al. found that the concentration gradients could be tailor-made [103] (as shown in Figure 8B) for a rational fabrication of a superstructure gradient along the radius in the sediment after the PUC, as shown in Figure 13B. Therefore, AUC is proven to be an important tool for the tailor-made concentration gradients, which helps with the formation of different binary nanoparticle-ordered structures in PUC. Moreover, this protocol may even be useful for the discovery of a library of crystalline phases for other species, such as a metal organic framework (MOF) [147]. For example, Park et al. [148] used reaction diffusion (RD) systems [149] to obtain the concentration gradients of both coordinating metal ions and organic ligands in a gel, which finally led to several bands of different microcrystals at different positions (which also evolved with time), as shown in Figure 13C. A final piece of work, which used PUC to order binary nanoparticles, was shown by Song et al. [150] They ordered binary gold nanoparticles of two different sizes into hetero clusters of different sizes by “crashing” one type of gold nanoparticles into a layer of the other type under the centrifugal force in PUC. To sum up, binary nanoparticles can be rationally ordered into a variety of superstructures in PUC with the sufficient knowledge from AUC experiments beforehand.

### 3.4. Highlight of Gradient Materials

A unique application of the PUC is the fabrication of gradient materials [152], which have a gradient transition in microstructure features instead of a sharp boundary. The application of centrifugation in making gradient materials is summarized in several books [153,154] and reviews [155,156]. We already showed in Figure 9 that nanoparticle gradient materials can be achieved in the PUC for monodisperse nanoparticles. This can be extended to more complicated binary nanoparticle mixtures. Chen et al. [157] showed that the latex nanoparticles of two different sizes can be ordered into gradient materials in the PUC, with the concentration of smaller nanoparticles gradually decreasing and that of larger nanoparticles gradually increasing along the radius, as shown in Figure 14. It was further shown that this method is even applicable to ternary nanoparticle mixtures [157] or a mixture of nanoparticles of completely different chemical natures, such as the mixture of latex and metal organic framework (MOF) nanoparticles [158]. Moreover, a functional gradient can be added into this structural gradient to make a double graded material, as shown by Bahner et al. [159] They modified latex nanoparticles of one size with silver patches and then mixed these functionalized nanoparticles with bare latex nanoparticles of a different size. Then the sediment after the PUC featured both structure and chemical gradients. These nanoparticle gradient materials indeed represent a new class of material, which may be easily fabricated in the PUC in a controllable fashion.

### 3.5. Current Status and Future Possibilities

We demonstrated in this chapter that ordering of nanoparticles in the PUC has shown a series of promising results in recent decades. Dating back to the 1990s, the ordering of monodisperse nanoparticles was conducted in the PUC only empirically, because nanoparticles with a narrow size and shape distribution crystallize very fast [160], which is not sensitive at all to nanoparticle concentration and centrifugal force. The same story holds for glassy superstructures, since the presence of oligomers suppresses the crystal nucleation and growth significantly [27,89]. Nevertheless, we showed that with the assistance of the AUC, the tuning of nanoparticle ordering can be rational, which means that AUC can provide important information about the amount of oligomers in the dispersion during centrifugation [144], which finally determines whether a crystal or glass may form. Therefore, we have a handle to tailor-make superstructures with different degrees of order by systematically changing the ionic strength of the dispersion and the centrifugation force. The importance of AUC becomes more evident when multi-modal nanoparticles are ordered in the PUC because the final superstructures after centrifugation are heavily dependent on so many different factors that the in-situ characterization of nanoparticles in the AUC becomes vital. We thus showed that with the sedimentation information from AUC, different binary superstructures can be realized and tuned in the PUC due to tailor-made concentration gradients. The field of rational nanoparticle ordering in the PUC just emerged and we shall see more developments in the future with the following aspects: (1) software development may be needed by which we can design tailor-made concentration gradients in a convenient way for the fabrication of desired superstructures in PUC experiments; (2) the ordering of nanoparticles may be realized on an industrial scale, which will increase the impact beyond academics; (3) the ordering of a mixture of nanoparticles with complicated shapes and chemical compositions can be targeted if multi-functionality is desired; (4) templating or confinement can be combined with the centrifugation for a more controllable ordering process; (5) the measurement of concentration gradients without the need of fluorescence labelling and refractive index matching, especially at high concentrations, will improve the versatility of this approach.

## 4. Post-Fabrication after Preparative (Ultra)Centrifugation (PUC)

After the PUC experiments, the sediment is phase-separated from the supernatant in the suspension. The next nontrivial technical step is to collect the sediment part out of the suspension without the disturbance of the superstructure inside, which determines the functional properties of the final materials [161]. Four main processing methods are normally employed to transform the sediment after the PUC to final functional materials. Firstly, the most intuitive method is evaporation. To avoid pronounced out-of-equilibrium effects [162] during the evaporation, such as fluid flows [163], solvent fluctuations [164] and capillary forces [165], a controlled and slow evaporation method may be applied. As Chen et al. showed [107], the remaining supernatant at the top of a centrifuge tube was removed as much as possible slowly and then the sediment was placed carefully into a sealed desiccator over silica gel for drying. Pressure, humidity and temperature can be also adjusted for a proper evaporation environment [166,167] so that the out-of-equilibrium effects may be minimized. Secondly, a freeze-drying protocol [168,169], which is well known for improving long term nanoparticle stability [170], can be also employed to dry the sediment without any effect from the capillary force. Freeze drying can preserve the nanoparticle ordering almost perfectly despite macropores due to ice formation in the freezing process. Thirdly, hydrogel trapping [171,172,173] is recently developed to keep the superstructure in place after centrifugation. For example, resorcinol-formaldehyde (RF) gel [157,174] and poly-(ethylene glycol) diacrylate (PEGDA) gel [175] were both successfully used to preserve the nanoparticle ordering after the PUC experiments. Last but not least, the infiltration and following removal of template nanoparticles [176,177] by means of sintering and etching is a common method to convert ordered nanoparticle structures to mesoporous or macroporous materials [178,179]. The presence of meso or macropores provides much room for a lot of applications [180]. As far as we realize, these four methods are commonly applied to fabricate final functional materials for a variety of applications.

## 5. Applications

After a proper post-fabrication, the materials can be finally prepared for different applications. Three main categories of applications are briefly discussed in this section: (1) crystals and glasses; (2) porous materials; and (3) functional gradient materials. The range of different applications is summarized in Table 1.

### 5.1. Crystals and Glasses

The most direct application of these materials relies on the structural order of nanoparticles after the centrifugation. Generally speaking, the nanoparticle ordering can be classified into two categories: a crystal, which is periodically ordered across a long range [181], and a glass, which lacks a long-range order [182]. The most promising application of these ordered/disordered structures is the utilization of their photonic properties [183]. Photonic crystals [139,184,185] have been studied for more than a decade so far to control and manipulate light [186,187]. Of particular interest is the photonic bandgap of a photonic crystal. In this bandgap, within a specific range of frequencies, light is forbidden to exist inside the crystal. However, any defect existence in the otherwise perfect crystal will lead to localized photonic states, which can be used to “mold” or control the properties of light, as shown in Figure 15A. A very promising example, which attracted much attention recently, was presented by Hynninen et al. [188] They found that by constructing crystal superstructures of a MgCu_2_ Laves phase [189] from binary nanoparticles, a bandgap in the visible region can be achieved, which may bring about many exciting possibilities in the future. In the meantime, disordered photonics [190] also attract much research interest. Due to the random packing of nanoparticles in the materials, light is multiply scattered [191,192] and leads to many disorder-based optical applications. For example, Garcia et al. [193] demonstrated the resonant behavior (such as in diffusion constant and energy velocity of light) for photonic glasses, as shown in Figure 15B. This study opens a new route to tune the light diffusion by employing photonic glasses.

### 5.2. Porous Materials

After the infiltration and removal of templating nanoparticles, porous materials [195] can be fabricated. There is a large number of reviews [179,196,197] on the applications of macro- (pore diameters larger than 50 nm) and meso- (pore diameters between 2 and 50 nm) porous materials [198]. Most applications arise from the presence of the pore structure, which provides an extremely large surface area over a relatively small volume. This makes porous materials suitable for many applications, such as catalysis, chemical sensing, selective adsorption, and molecular separation. For example, Arandiyan et al. [194] demonstrated that Pt nanoparticles embedded in macroporous materials made of Ce_0.6_Zr_0.3_Y_0.1_O_2_ (3DOM CZY) possessed an excellent efficiency for the catalysts of methane combustion, as shown in Figure 15C.

### 5.3. Functional Gradient Materials

Functional gradient materials [199] exhibit spatial gradients of microstructures and properties [152,156,200]. Humans have extensively used these materials since the old ages of craftsmanship and engineering constructions, especially those like case-hardened steel, which is still commonly used today [152]. The microstructure gradient can be quantitatively controlled for the optimization of material properties for specific requirements, such as mechanical [201], thermal [154] and chemical properties [200]. The related applications [202] range into different industrial fields, such as aerospace, defense, mining, power and tool manufacturing sectors. Especially, nanoparticle gradient materials have recently proven to be rather feasible to tailor-make by employing the centrifugation technique [96]. Chen et al. [157] demonstrated that binary nanoparticle gradients such as a mixture of latex nanoparticles of different sizes were fabricated by centrifugation. The resulting gradient material was then infiltrated with resorcinol-formaldehyde. The thermal treatment finally induced the transformation to gradient porous materials. The final material can be applied as an electrode in a Li-O_2_ cell [203], as shown in Figure 15D. The cell performance can be easily tuned by both the gradient structure and the gradient direction.

## 6. Summary

The sedimentation of microbeads [204,205] in gravity saw many theoretical and experimental advances in the last century. With the emergence of nanotechnology [206,207], the ordering of nanoparticles by sedimentation attracts a lot of attention, which requires the employment of a centrifugal field [208]. The ordering of monodisperse nanoparticles represents an intuitive and simple situation. Colloidal crystals and glasses were both fabricated successfully in preparative (ultra)centrifugation [67]. However, when the composition of nanoparticle mixtures becomes complex, the ordered superstructures become obscure, which thus requires an in-situ observation and better understanding of the ordering process in a centrifugal field. Analytical (ultra)centrifugation methods [209], which include sedimentation velocity and sedimentation-diffusion equilibrium experiments, can precisely monitor the ordering process of nanoparticles in real time, especially for a multiple particle mixture using a multiwavelength UV-Vis absorbance detector if the refractive index of the nanoparticles is matched by the solvent to avoid turbidity by light scattering. An important achievement so far in this field is the fabrication of tailor-made concentration gradients with the pre-knowledge of sedimentation and diffusion properties of the nanoparticles. The successful transition from analytical to preparative centrifugation furthermore enhanced the accessibility and applicability of this technique. Besides conventional purification and separation [123], preparative centrifugation can also be employed to order nanoparticles in a rational and controllable manner. In this case, complicated superstructures can be formed with full tuneability. After a proper post-fabrication of these superstructures, functional materials can be thus prepared, which showed promising applications in different fields, especially as photonic, porous and gradient materials. The centrifugation technique has already shown many promising results, both in fundamental and application aspects of the ordering of nanoparticles. As far as we realize, there are two main challenges in this area: (1) software development to make tailor-made superstructures more convenient to design, with the combined use of analytical and preparative (ultra)centrifugation. By filling this gap, a customized and easy-to-use strategy is possible for researchers to make any desired gradient superstructures for different applications; (2) methodology development to allow the measurement of concentration gradients of nanoparticles in centrifugation without the pre-requirement of a refractive index matching solvent and fluorescence labelling, especially at high concentrations. This will certainly improve the application to all kinds of nanoparticles, besides the well-studied silica and latex nanoparticles. By solving this problem, a large range of concentrations for all kinds of materials will be available for researchers to make nanoparticle-based materials of unique properties. Therefore, we expect to see more exciting research output on (ultra)centrifugation techniques in the area of nanoparticles in the future.

## Figures and Tables

**Figure 1 nanomaterials-11-00333-f001:**
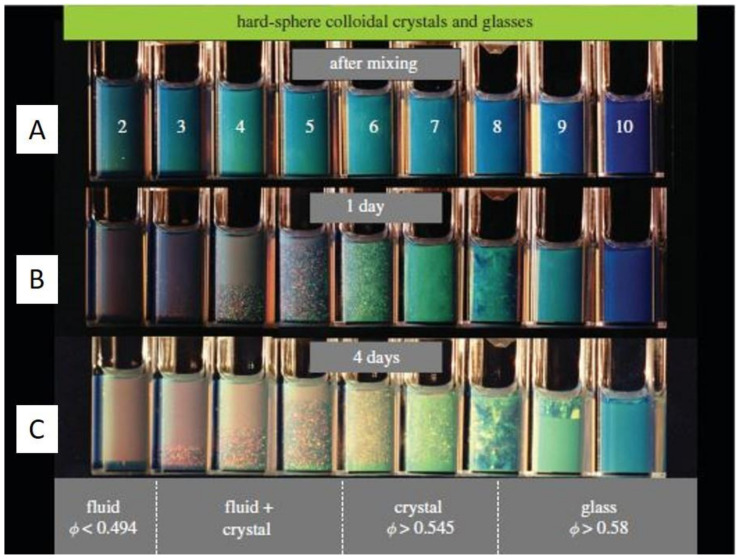
Hard-sphere latex suspensions in cuvettes, illuminated obliquely from behind by white light, (**A**) immediately after mixing, (**B**) after 1 day, and (**C**) after 4 days. In the lowest column (after 4 days), the lower part of the right cuvettes shows the glassy phase with the volume fraction larger than 0.58, the middle cuvettes show the nucleated crystal phase with the volume fraction between 0.545 and 0.58 while the left cuvettes show mostly the fluid phase when the volume fraction is lower than 0.494. Reproduced with permission from [18], Copyright the Royal Society, 2009.

**Figure 2 nanomaterials-11-00333-f002:**
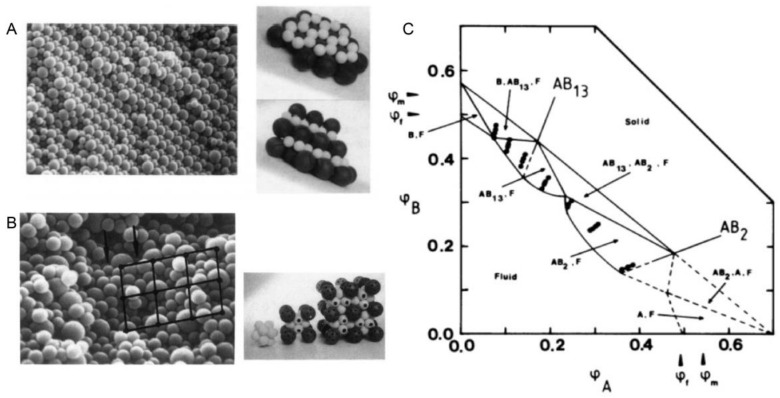
(**A**). Scanning electron micrograph and model of an AB_2_ superlattice of the sample composition: *n*_B_/*n*_A_ = 6, *φ*_A_ + *φ*_B_ = 0.536 for a binary hard sphere mixture at size ratio *R*_B_/*R*_A_ = 0.58; (**B**). Scanning electron micrograph and model of an AB_13_ superlattice of the sample composition: *n*_B_/*n*_A_ = 9, *φ*_A_ + *φ*_B_ = 0.552 for a binary hard sphere mixture at the same size ratio; (**C**). Phase-diagram of a binary hard sphere mixture at size ratio *R*_B_/*R*_A_ = 0.58. The axes are the volume fractions *φ*_A_ of particle A and *φ*_B_ of particle B. *φ*_f_ and *φ*_m_ indicate the freezing and melting concentrations of each species. The points show the samples studied. At the low concentration, the phase is fluid (*F*), while at higher concentrations four regions of coexistence of crystal (*B*, *AB*_13_, *AB*_2_, or *A*) and fluid (*F*) are found, where *B* or *A* denotes the crystal phase of only one species of *B* or *A*, *AB*_2_ the atomic analogue of borides such as AlB_2_ and *AB*_13_ the atomic analogue of NaZn_13_, UBe_13_, etc. Reproduced with permission from [24], Copyright the American Physical Society, 1992.

**Figure 3 nanomaterials-11-00333-f003:**
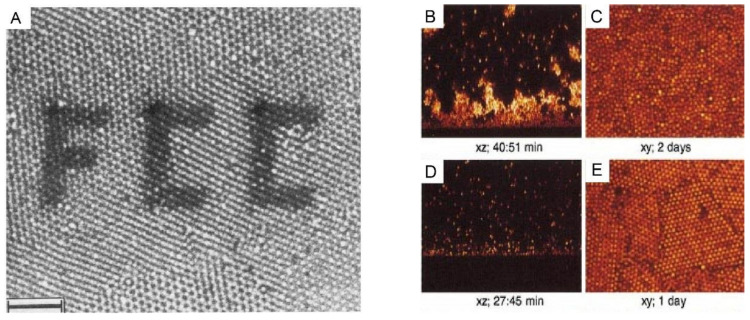
(**A**): A polycrystalline section of 1-octadecanol-coated silica spheres (particle radius 505 nm, initial volume fraction 20%) in chloroform inside the capillary, observed by confocal laser scanning microscope (CLSM). The photobleached pattern (FCC) was created by the high-intensity illumination with the imaging beam. Scale bar: 10 µm. (**B**–**E**): CSLM images of xz scans after an hour and of xy scans after several days for the sedimentation of polymethylmethacrylate (PMMA) particles in a reactive index matching solvent of (**B**,**C**): initial volume fraction: 2.05%, polymer concentration: 51.72 mg/mL; and (**D**,**E**): initial volume fraction: 2.18%, polymer concentration: 51.18 mg/mL. The polymers were used here to tune the interaction strength between these colloidal particles. Reproduced with the permission from [34], Copyright the American Chemistry Society, 1992; and [33], Copyright the American Physical Society, 2001.

**Figure 4 nanomaterials-11-00333-f004:**
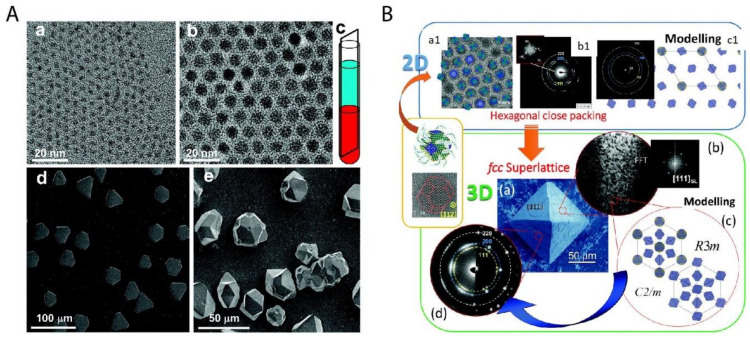
Two important examples of superstructures from the bottom-up organization of nanoparticles. (**A**) The colloidal superlattices made of spherical PbS nanoparticles of 3.1 nm (**a**) and 8.0 nm (**b**) are shown in (**d**) and (**e**) respectively; (**c**) shows the schematic fabrication approach of the oversaturation technique for growing superlattices. (**B**) The piece of mesocrystal, as shown in the green box: (**a**) is the SEM image of the mesocrystal with (**b**) the corresponding Fast Fourier transform (FFT) of one Focused ion beam (FIB) slice, (**c**) the simulated structural models of the mesocrystal, and (**d**) the experimental and simulated electron diffraction (ED) pattern. The mesocrystal was synthesized by the bottom-up fabrication from truncated octahedrally shaped PbS nanoparticles of 5.5 nm, as shown in upper left (**a1**) with (**b1**) the corresponding ED pattern of the 2D assembly and (**c1**) the structural modeling of the assembly in the blue box. Reproduced with permission from [45], Copyright the American Chemical Society, 2010; and [47] Copyright the Royal Society of Chemistry, 2016.

**Figure 5 nanomaterials-11-00333-f005:**
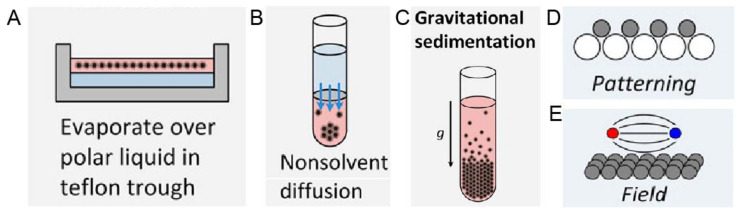
Schematic experimental approaches to prepare superstructures from nanoparticles, including (**A**) Solvent evaporation based methods, such as evaporation over a polar liquid in a Teflon trough; (**B**) Solvent destabilization-based methods, such as nonsolvent diffusion; (**C**) Gravitational sedimentation, which is a less common method due to insignificant gravitational forces for intrinsically small sizes of nanoparticles; (**D**) Pattering, where the template is normally used to guide the ordering and (**E**) Ordering in an external field, especially magnetic and electric fields. Reproduced with permission from [55], Copyright the American Chemical Society, 2016.

**Figure 6 nanomaterials-11-00333-f006:**
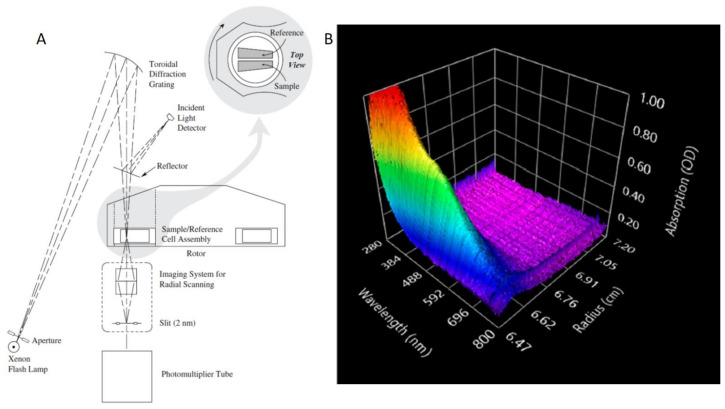
(**A**) Schematic setup of a Beckman CoulterTM Optima^TM^ XL-A Analytical UltraCentrifugation (AUC) instrument (with an UV absorption optic system). Reproduced with permission from [86], Copyright Beckman-Coulter, 1993. (**B**) Typical three-dimensional dataset result for an AUC experiment with a multi-wavelength (MWL) optical system.

**Figure 7 nanomaterials-11-00333-f007:**
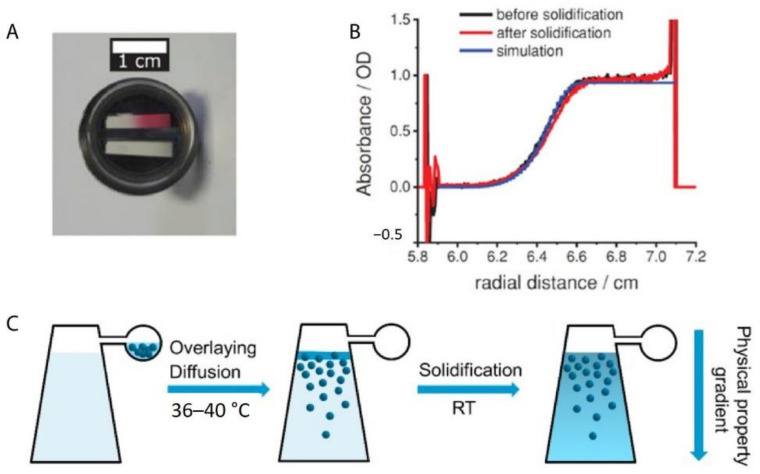
(**A**) Photograph of a piece of gradient nanoparticle superstructure made of spherical gold nanoparticles in gelatin inside an AUC centerpiece. The top sector is the sample sector with the nanoparticle gradient, while the bottom sector is the reference without any nanoparticles inside; (**B**) Absorbance profile of the gold nanoparticle concentration gradient, obtained after the centrifugation at 13,000 rpm for 32 min. The experimental gradient did not change after the solidification and agrees with the simulation results with the SEDFIT; (**C**) Schematic procedures to produce nanoparticle gradient superstructures in the sample sector of an analytical ultracentrifugation band-forming cell by the diffusion at a low angular velocity and subsequent solidification. Reproduced with permission from [98], Copyright Wiley-VCH, 2018.

**Figure 8 nanomaterials-11-00333-f008:**
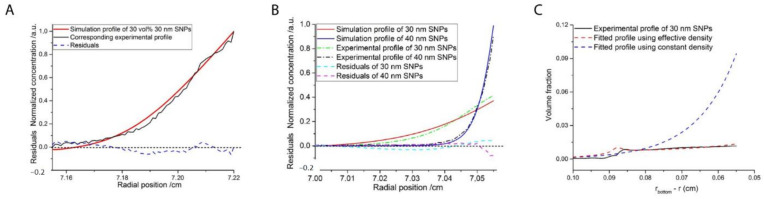
(**A**) Normalized experimental (black line) and simulation (red line) concentration (*c*) gradients for 30 nm hard-sphere-like silica nanoparticles of an initial concentration of 30 vol% in an AUC-SE experiment at 2800 rpm; the root-mean-square deviation (RMSD) value is 0.034% and the blue line corresponds to the deviation between experimental and simulation data; (**B**) Normalized experimental (dash dotted lines) and simulation (solid lines) concentration gradients for a binary hard sphere mixture of 30 nm (3.3 vol%) and 40 nm (6.7 vol%) silica nanoparticles in an AUC-SE experiment at 2800 rpm and the dashed line corresponds to the deviation between experimental and simulation data; (**C**) Experimental (solid line) and simulation (dashed lines) concentration gradients for 30 nm silica nanoparticles in a binary hard sphere mixture of 30 nm (1 vol%) and 90 nm (9 vol%) silica nanoparticles in an AUC-SE experiment at 2800 rpm, with the simulation profiles using the constant solvent density (blue dashed line) and the effective density (red dashed line); Reproduced with permission from [103], Copyright the American Chemical Society, 2019, and [106], Copyright Wiley-VCH, 2019.

**Figure 9 nanomaterials-11-00333-f009:**
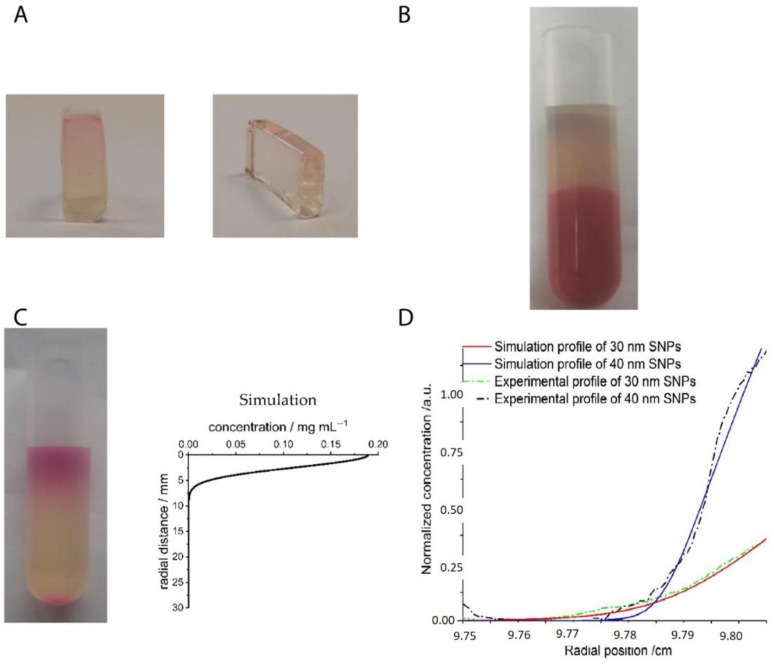
(**A**) A typical piece of nanoparticle gradient material in a gelatin gel matrix made in an AUC cell from two different views with the dimension of 1.2 cm × 0.7 cm × 0.3 cm; (**B**) Typical piece of nanoparticle gradient material made in a cylindrical PUC tube with the length of 3.8 cm and diameter of 1.2 cm; (**C**) Piece of fluorescence labeled silica nanoparticle gradient material obtained after a PUC experiment at 3000 rpm for 20 h in a L-70 preparative ultracentrifuge with the corresponding simulated concentration gradient; (**D**) Normalized experimental (dashed dotted lines) and simulation (solid lines) concentration gradients for a binary hard sphere suspension of 30 nm (3.3 vol%) and 40 nm (6.7 vol%) silica nanoparticles with two different fluorescent labels from a PUC experiment at 2800 rpm in a L-70 preparative ultracentrifuge. The absorbance gradients were measured by a confocal laser scanning microscope (CLSM) and agreed with the simulations. Reproduced with permission from [98], Copyright Wiley-VCH, 2018, and [103], Copyright the American Chemical Society, 2019.

**Figure 10 nanomaterials-11-00333-f010:**
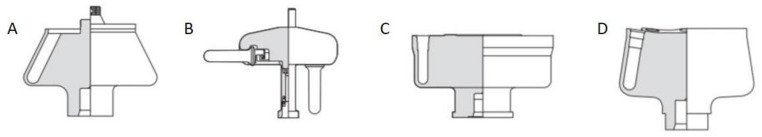
Four main types of rotors used in the preparative ultracentrifugation experiments, including (**A**) fixed-angle rotor, (**B**) swinging-bucket rotor, (**C**) vertical rotor and (**D**) near-vertical rotor. For the ordering of nanoparticles, a swinging-bucket rotor is recommended since all the nanoparticles can sediment horizontally towards the bottom of the tube, which is most comparable to an AUC cell. Reproduced with permission from [86], Copyright Beckman-Coulter, 1993.

**Figure 11 nanomaterials-11-00333-f011:**
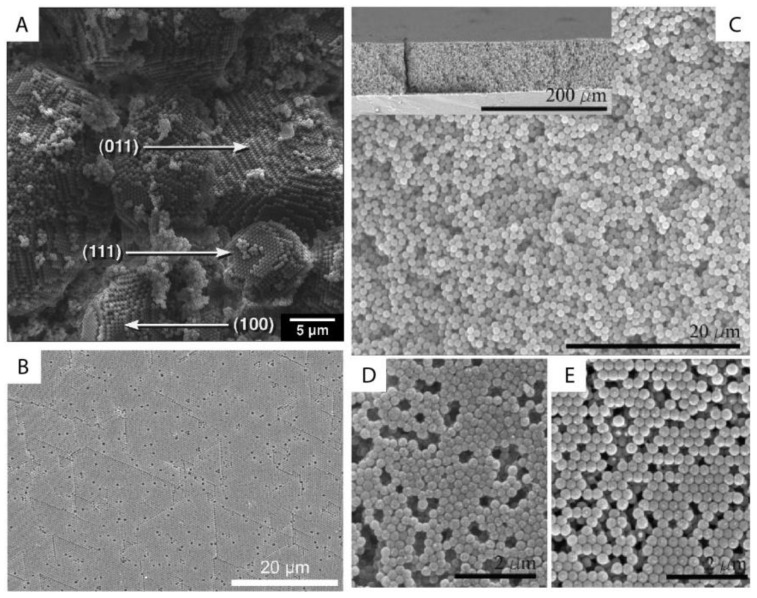
(**A**) SEM image of a small colloidal crystal of monodisperse latex nanoparticles after the centrifugation with (100), (111) and (011) faces being labelled; (**B**) SEM image of a crystalline film of monodisperse silica nanoparticles after the centrifugation experiment; (**C**–**E**) SEM image of a colloidal glass of latex nanoparticles after the centrifugation experiments. In (**C**), salt was added to break the stability of these particles and to introduce the coagulation for the disordered structure. In (**D**,**E**), a second type of nanoparticle was introduced first and then selectively etched for the disordered structure. Reproduced with permission from [133], Copyright the American Chemical Society, 1999, and [141], Copyright Wiley-VCH, 2007.

**Figure 12 nanomaterials-11-00333-f012:**
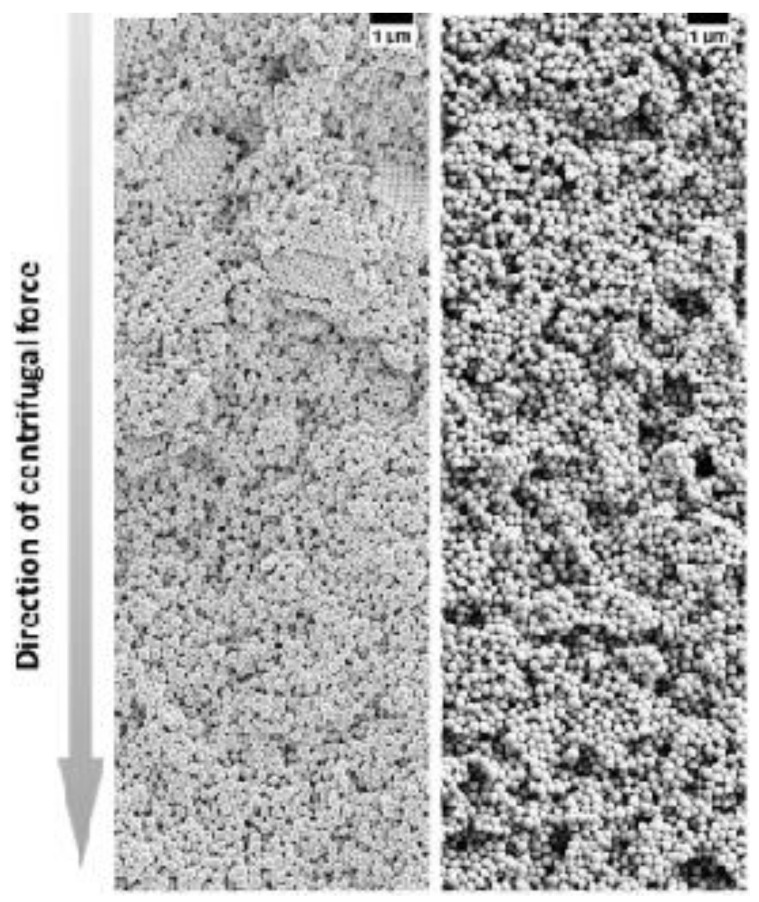
SEM images of colloidal glasses prepared in centrifugation in the presence of different salt concentrations. With a lower salt concentration, small crystalline domains can be still present in the upper part of the sediment (left panel) while at a higher salt concentration, the superstructure becomes totally disordered (right panel). Reproduced with permission from [144], Copyright Wiley-VCH, 2017.

**Figure 13 nanomaterials-11-00333-f013:**
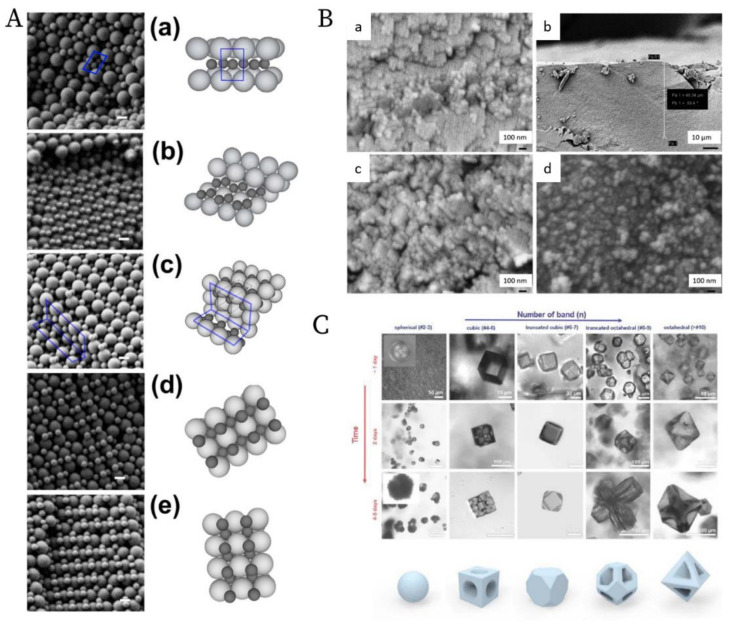
(**A**). SEM images and schematics of the corresponding binary crystal lattices at different radial positions for a binary mixture of 150 nm and 300 nm latex nanoparticles after the PUC: (**a**) layered structure akin to AlB2; (**b**) distorted AlB2 structure; (**c**) the tetragonal A3B variant; the tetragonal A2B variant before (**d**) and after distortion (**e**). Scale bars = 200 nm; (**B**). SEM images at different radial positions for a binary mixture of 30 nm and 40 nm silica nanoparticles after the PUC: (**a**) Representative image of a crystalline structure in the upper part of the sediment; (**b**) Overview of the transition position where the last piece of crystalline structure was observed (≈50 µm from the meniscus); (**c**) Representative zoom-in image of the transition position where only discrete small pieces of crystalline structures were observed. (**d**) Representative image of a glassy structure below the transition position; The crystal to glass transition agreed with the previous study [151]; (**C**). Confocal microscopy images of the morphology evolution of the MOF-crystal in the precipitation bands at different positions (horizontal axis) due to the concentration gradients of both metal ions and ligands, and at different time (vertical axis). Reproduced with permission from [107], Copyright the American Chemical Society, 2015, [103], Copyright the American Chemical Society, 2019 and [148], Copyright Wiley-VCH, 2020.

**Figure 14 nanomaterials-11-00333-f014:**
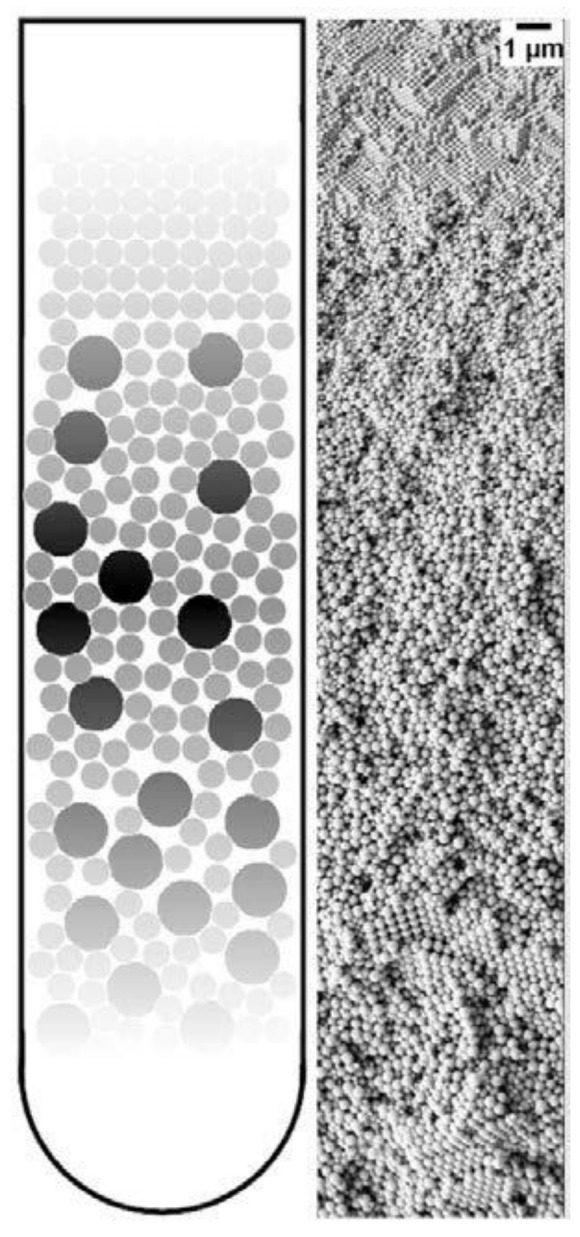
Schematic image of a nanoparticle gradient material, made of a binary mixture of latex nanoparticles (**left**) and a representative SEM image of this gradient structure (**right**). Reproduced with permission from [107], Copyright the American Chemical Society, 2015.

**Figure 15 nanomaterials-11-00333-f015:**
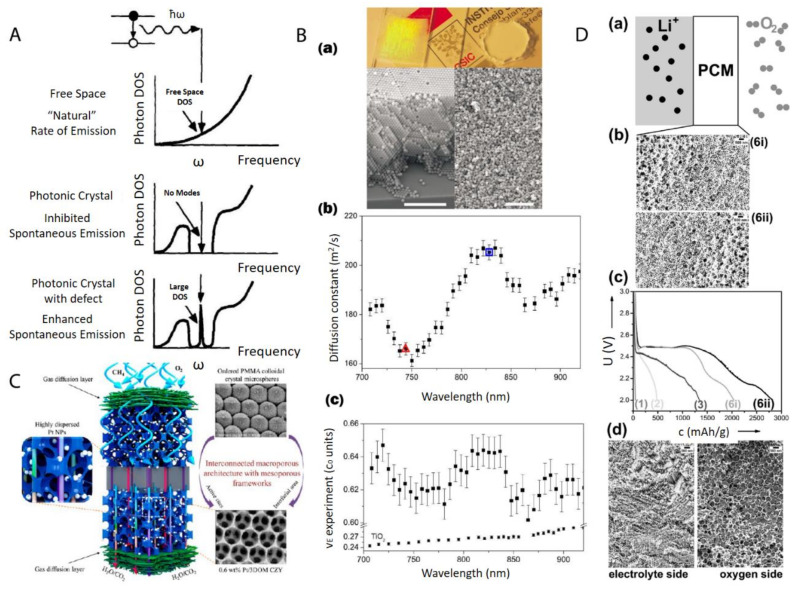
(**A**) Photon density of states (DOS) in free space, a photonic crystal, and a photonic crystal with a local defect. A perfect photonic crystal inhibits spontaneous photon emission within a band gap centered at a frequency *ω* while the spontaneous photon emission can be enhanced dramatically at frequency *ω* by a localized point defect in the colloidal crystal; (**B**) (**a**) Top: picture shows a photonic crystal (left), which shows visible iridescences and a photonic glass (right) white and without any iridescences. Bottom: Left is the SEM image from a photonic crystal (scale bar is 10 µm) while Right is the SEM image from a photonic glass (scale bar is 10 µm); (**b**) Experimental diffusion constant in a photonic glass made of PS spheres. Two particular wavelengths are marked with a triangle (*λ*_1_ = 744 nm) and a square (*λ*_2_ = 828 nm). These two wavelengths correspond to the maximum and minimum value of a Mie resonance; (**c**) Experimental values of the energy velocity (*ν*_E_) as a function of the wavelength (*λ*) for a photonic glass made of PS spheres and for TiO_2_ powder. For the photonic glass, a full oscillation of around 5% amplitude is illustrated, while for the TiO_2_ powder, a flat velocity dispersion is illustrated; (**C**) Schematic picture of a highly efficient catalysis system for methane combustion, which is composed of Pt nanoparticles embedded in latex nanoparticle-templated macroporous structures; (**D**) (**a**,**b**) porous gradient material-based Li-O_2_ battery testing setup; (**c**) different Li-O_2_ electrochemical discharge curves by using different gradient pore sizes; (**d**) SEM image of a typical porous gradient structure after charging/cycling. Reproduced with permission from [186], Copyright Elsevier, 1997; [193], Copyright the American Physical Society, 2008; [194], Copyright the American Chemical Society, 2015; and [157], Copyright Wiley-VCH, 2017.

**Table 1 nanomaterials-11-00333-t001:** Examples of different applications for the material fabricated by (ultra)centrifugation for nanoparticles.

Application Category	Application	Reference
crystal and glass	light manipulation	Pearson, 1972; Garcia, et al., 2008
porous material	catalysis	Arandiyan, et al., 2015
	chemical sensing	Stein, et al., 2001
	selective adsorption	Cecilia, et al., 2019
	molecular separation	Cecilia, et al., 2019
	fuel cell	Li, et al., 2016
functional gradient material	fuel cell	Holtappels, et al., 2006
	coating	Saleh, et al., 2020
	biomedical sector	Bhavar, et al., 2017

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
