# Peer review of "Ultracentrifugation Techniques for the Ordering of Nanoparticles"

_nanomaterials, 2021, doi:10.3390/nano11020333_

Round 1

Reviewer 1 Report

Xu and Cölfen provide a comprehensive overview of AUC's past, present, and future. To be honest, this manuscript is a nicely written review and quite an informative read. I particularly enjoyed the first two sections on microbeads and nanoparticles in gravity, but overall there is little room for criticism. It is a recommended read for both students of colloidal and physical-chemical sciences, but also for researchers in other fields. I usually tend to provide an extensive list of comments and questions, but in this case I am almost completely satisfied with the state of the manuscript. One point that the authors might want to elaborate on is the aspect of AUC as a tool for separating colloidal assemblies of nanoparticles. Colloidal superstructures in the form of clusters of gold nanoparticles as well as hierarchical assemblies, e.g., with core/satellite morphology, tend to exhibit challenges in terms of their separation from their non-assembled building blocks and other formed assembly species. Pazos-Perez et al. presented one such case in which assemblies with different coordination numbers were successfully separated (10.1002/anie.201207019).  Given the potential outlined, AUC could also be a valuable tool for the preparation (and optical/structural characterization) of more complex colloidal superstructures with isotropic (10.1021/acsami.0c16398) and especially anisotropic building blocks (nanorods: 10.1039/c9nr06102a, triangular cores: 10.1021/acsphotonics.0c00642) as reported by Höller et al. These are the first that come to mind, but I'm sure other examples can be found. However, to my knowledge, AUC has not yet been applied to such anisotropic superstructures. In addition, I noticed a typo (p.5 l.148 "filed") and a typographical error (l.191 T should be italicized). Also, the authors might consider enlarging the panels shown in Fig. 8, which are barely legible.

Overall, I fully recommend publication of this excellent review in Nanomaterials after minor revisions as indicated above.

Reviewer 2 Report

This work revises the state of the art on the use of ultracentrifugation, both for the production of nanocrystals to its use to obtain concentration gradients for advanced materials. The topic is worth of investigation and the state of the art is indeed very well discussed. Among the cited 204 references however only two are dated from 2020 (and from these one seems to be a book). A more up-to-date use of references would be recommended because the use of centrifugal field for the production of gradient materials is not an old topic.  The title does not reflect the full content of this review, while section 5 is only briefly outlining the applications. Maybe a table illustrating a range of applications would be suitable.

Reviewer 3 Report

Excellent review of technique for nanoparticle ordering studies over multiple decades. The description of this area is comprehensive and understandable to the non-specialist in analytical and preparative ultracentrifugation techniques. The authors make a convincing case for the use of analytical ultracentrifugation (AUC) as a tool to guide in the application of defined ordered nanoparticle superstructures using preparative ultracentrifugation (PUC).

One minor correction is suggested to the caption of Figure 4. The abbreviations FFT, FIB and ED should be stated in parenthesis after stating the full definition of the abbreviation, for example, electron diffraction (ED). This will be helpful to non-specialist in understanding the information being conveyed in the figure's caption.

Reviewer 4 Report

Overall, it's a well-written manuscript summarizing nanoparticles for the centrifugal field. Here are some comments. 1. More depth of discussion is desired in the conclusion, e.g. what are future steps or challenges? 2. Title is a bit misleading, it looks like a shopping guide instead of a scientific review.
